# Extraction of Forest Road Information from CubeSat Imagery Using Convolutional Neural Networks

Lukas Winiwarter [1,2,3], Nicholas C. Coops [1,*], Alex Bastyr [1], Jean-Romain Roussel [4], Daisy Q. R. Zhao [1], Clayton T. Lamb [5] and Adam T. Ford [5]

1 Integrated Remote Sensing Studio, Faculty of Forestry, University of British Columbia, Vancouver Campus, Vancouver, BC V6T 1Z4, Canada; lukas.winiwarter@uibk.ac.at (L.W.); zhaoqinrui@gmail.com (D.Q.R.Z.)
2 Department of Geodesy and Geoinformation, TU Wien, 1040 Vienna, Austria
3 Unit of Geometry and Surveying, Faculty of Engineering Sciences, University of Innsbruck, 6020 Innsbruck, Austria
4 Centre de Recherche sur les Matériaux Renouvelables, Département des Sciences du Bois et de la Forêt, Université Laval, Québec, QC G1V 0A6, Canada; jean-romain.roussel.1@ulaval.ca
5 Department of Biology, University of British Columbia, Okanagan Campus, Kelowna, BC V1V 1V7, Canada; clayton.lamb@ubc.ca (C.T.L.); adam.ford@ubc.ca (A.T.F.)
* Correspondence: nicholas.coops@ubc.ca

**Abstract:** Forest roads provide access to remote wooded areas, serving as key transportation routes and contributing to human impact on the local environment. However, large animals, such as bears (*Ursus* sp.), moose (*Alces alces*), and caribou (*Rangifer tarandus caribou*), are affected by their presence. Many publicly available road layers are outdated or inaccurate, making the assessment of landscape objectives difficult. To address these gaps in road location data, we employ CubeSat Imagery from the Planet constellation to predict the occurrence of road probabilities using a SegNet Convolutional Neural Network. Our research examines the potential of a pre-trained neural network (VGG-16 trained on ImageNet) transferred to the remote sensing domain. The classification is refined through post-processing, which considers spatial misalignment and road width variability. On a withheld test subset, we achieve an overall accuracy of 99.1%, a precision of 76.1%, and a recall of 91.2% (F1-Score: 83.0%) after considering these effects. We investigate the performance with respect to canopy coverage using a spectral greenness index, topography (slope and aspect), and land cover metrics. Results found that predictions are best in flat areas, with low to medium canopy coverage, and in the forest (coniferous and deciduous) land cover classes. The results are vectorized into a drivable road network, allowing for vector-based routing and coverage analyses. Our approach digitized 14,359 km of roads in a 23,500 km² area in British Columbia, Canada. Compared to a governmental dataset, our method missed 10,869 km but detected an additional 5774 km of roads connected to the network. Finally, we use the detected road locations to investigate road age by accessing an archive of Landsat data, allowing spatiotemporal modelling of road access to remote areas. This provides important information on the development of the road network over time and the calculation of impacts, such as cumulative effects on wildlife.

**Keywords:** SegNet; RGB imagery; Planet imagery; road network; vectorization

## 1. Introduction

The provision of up-to-date and spatially accurate information on the location of roads is increasingly fundamental for a range of conservation and industrial activities globally [1], and there is growing recognition that the ecological implications of roads can be far-reaching [2]. As a result, there is an emerging field of road ecology to improve our understanding of both the direct and indirect effects of roads on the environment, especially wildlife populations [3]. Roads are critical for the extraction of timber, minerals, fossil fuels, and arable land. New road construction is highly dynamic as infrastructure is developed

to access sites and transport materials. In a forestry context, roads are required for effective industrial forest management, providing access for harvesting and silvicultural operations, but also for recreational, wildlife, and wildfire management. Such forestry-associated roads may occur in remote areas, with complex topography, and widely varying landcover in the surroundings.

Roads can, however, have long-lasting and pervasive impacts on ecosystems. Road networks, through increases in compaction of road surfaces, which, in turn, reduces infiltration, can modify the hydrological cycle [4]. Roads can have large impacts on environmental integrity and wildlife health through habitat fragmentation and alterations of landscape structure, which can result in cascading ecological implications such as an increased risk of mortality [5]. Roads can also be a major cause of animal mortality, particularly for large species such as bears (*Ursus* sp.), moose (*Alces alces*), and elk (*Cervus eleaphus canadensis*), who use road networks for movement, but also for food, as roadside verges with more sunlight and water than within the forest canopy [6,7]. Roads facilitate motorized human access to remote areas, causing habitat disruption [6]. Finally, some predator species, such as wolves, can move more efficiently on roads, which can have impacts on predation-sensitive species such as caribou (*Rangifer tarandus caribou*) [8].

Mapping and monitoring forest roads can be difficult, and databases that contain road information can be challenging to build and maintain. These difficulties emerge because road networks are highly dynamic, with new roads developed and older roads decommissioned at a rate far faster than public databases are updated and ground-checked. The province of British Columbia (BC), Canada, offers a complex case study in road design, building, and maintenance of its extensive road network covering over 800,000 km [9], of which 74% is related to the forest industry. Much of the road network exists in remote areas of BC, and the case study also includes a range of environmental conditions, including highly variable terrain, climate, and land form features [10]. The BC government reports that approx. 150,000 km of roads are missing from their database, with more than half of the existing data being out of date.

In Canada, and in fact, globally, most provincial, state, or territory governments maintain vector information on the forest roads under their jurisdiction [11]. Historically, road locations and associated attributes have been mapped using aerial photography and manually delineated road features. Today, this information is often augmented with field information, GPS tracks, road planning documents, and local geographic information systems. As a result, road positional information can be quite accurate, assuming a well-established, large road with clear features [12]. In forested environments, the interpreter is limited in what can be observed underneath forest canopies, resulting in significant position errors.

Previous work on road extraction from remotely sensed data has predominantly focused on urban areas [13–15] or more recently using active remote sensing data such as airborne laser scanning [11]. Other work has focused on low-resolution imagery (tens of meters in resolution) [16], whereas the following publications worked with resolutions that are commercially not available from satellites.

Çalışkan and Sevim [17] presented an investigation of forest roads from 10 cm orthophotos using four different deep learning models. They found significant differences between the models, with overall accuracy ranging between 89.15% and 98.49% and Intersection over Union (IoU) ranging between 61.81% and 67.31%. Depending on the metric, different models performed better than the others.

Zhang and Hu [18] used a VGG-16-based approach to extract primary and secondary roads from high spatial resolution imagery (40 cm) of Canadian forests. While they preprocessed data using a Laplacian of Gaussian (LoG) operation to locate road candidates, they did not employ weights from a pre-trained neural network.

While approaches using coarser spatial resolution data acquired by satellites have been presented, they typically do not focus on forest roads. For example, Hormese and Saravanan [19] show how urban roads can be detected using the VGG-16 architecture. In a

recent contribution, Kelesakis et al. [20] demonstrated the necessity of rural road data for a range of applications under different deep learning solutions.

In this work, we complement this by presenting a method of automatically extracting forest road data from high-resolution (3 m) RGB satellite imagery, including a vectorization of the resulting road network. To achieve fast convergence, we use a pre-trained Convolutional Neural Network (CNN), which we transfer to segment remote sensing data. Furthermore, by using a randomly initialized decoder network in a U-Net configuration, we predict the class labels for all pixels of the input image simultaneously, further increasing convergence performance. We present a novel evaluation method by investigating the results in the raster domain, considering errors arising from imprecise geolocation and vector simplification in the reference data. Furthermore, we analyze the performance with respect to terrain features (slope, aspect), vegetation cover (Normalized Difference Vegetation Index, NDVI), and land cover class. Comparison with historic satellite imagery gives insights into road age, allowing us to analyze road development in the last 40 years, and adding semantic information to the derived roads. As a post-processing step, we vectorize the road network using the vectorization algorithm published by Roussel et al. [21], resulting in a topologically valid network layer as required for many further analyses.

## 2. Materials and Methods

### 2.1. Study Area

To speed up processing and enable clear visualizations, we processed and presented data on two spatial areas. The larger area is defined by a polygon outlining important caribou herd ranges (Moberly, Scott, Kennedy Siding, Burnt Pine, Quintette, and Narrway herds) and spans approximately 23,500 km$^2$ in total [22,23]. These two study areas also aim to achieve different goals: on the smaller subset, where test and validation data are available, road predictions are evaluated quantitatively, whereas on the larger area, the potential for updating an existing road network is evaluated. Furthermore, training of the deep learning model is only undertaken on the smaller area, which allows for the analysis of the generalization capabilities of the trained classifier when applied on the larger area.

On the smaller area, we defined a rectangular, axis-aligned grid of $5 \times 2$ tiles spanning an area of $71 \times 145$ km$^2$ (a total area of 10,295 km$^2$). Of these 10 tiles, eight were used for training, one for testing, and one for validation. The choice of the areas was such that the road density for each of the subsets was balanced. The eight tiles are shown in Figure 1 with the RGB information. The spatial split between training and test/validation areas ensures that spatial autocorrelation does not distort the results.

### 2.2. CubeSat Data

We utilized data acquired from the Dove Classic satellites from the PlanetScope CubeSat constellation, which consists of over 130 CubeSats in sun-synchronous orbit at altitudes of approximately 475 km [24]. The constellation of satellites is such that it provides near-daily image acquisition globally, at a ground-sampling distance (GSD) of approximately 3 m. Imagery from the constellation archives is available beginning in 2016 and is acquired in 4 spectral bands: blue (455–515 nm), green (500–590 nm), red (590–670 nm), and near-infrared (780–860 nm). Of these bands, we only used blue, green, and red, as preliminary experiments showed that including the near-infrared information did not result in significantly better results, but increased the training time of the CNN. We selected tiles from Planet's Basemap product (https://www.planet.com/products/basemap/, last access: 12 May 2023), which are mosaicked from multiple acquisitions within a calendar month. Where tiles for the month of August 2017 showed smoke or clouds, we manually replaced them with a cloud- and smoke-free mosaic from either July or September 2017. Using data from 2017 ensured that the satellite imagery and the reference data (cf. Section 2.4) represent the same temporal state of the roads.

Multiple satellite overpasses covered the larger area of interest. We used 23 of these—partially overlapping—overpasses and included data from 5 July 2017 until

4 September 2017. Two small areas and one longer strip in the western part of the area of interest were not covered by satellite imagery of sufficient quality (cloud- and smokeless) and hence resulted in data gaps.

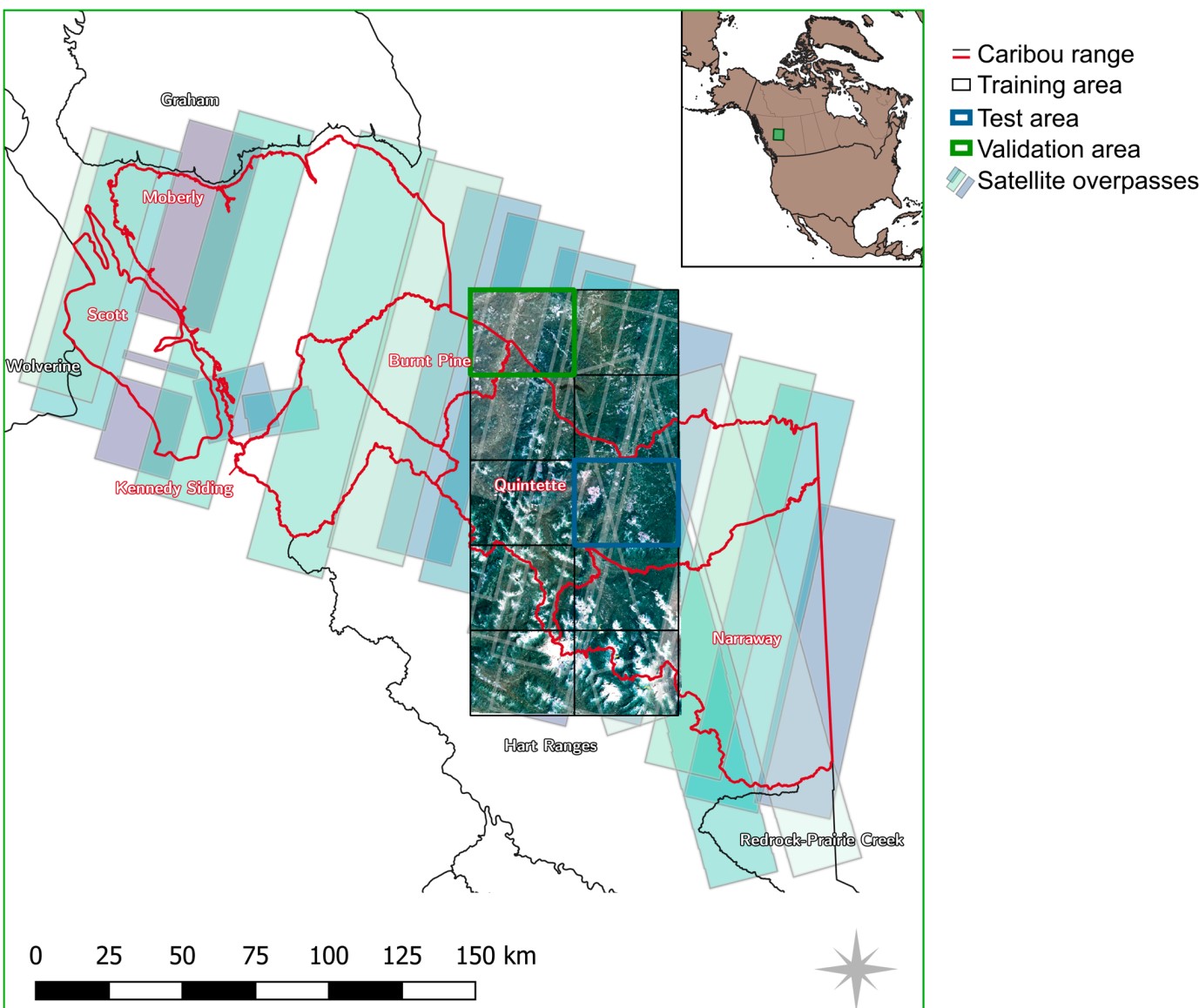

**Figure 1.** Overview map of the study area. The map shows two overlapping areas of interest: (i) 10 rectangular tiles showing the RGB composite satellite imagery, and (ii) the area covered by the satellite overpasses (in shades of green and blue). Additionally, Caribou herd ranges are marked in red, and the validation and test tiles are in green and blue, respectively. For training, the governmental dataset within the black rectangles is used. In the gaps in the north-west, no usable satellite data was found during the timespan of interest.

### 2.3. Landsat Historical Information for Road Age

We further utilized annual Landsat image composites generated through the Composite2Change (C2C) approach [25] to conduct a temporal analysis for determine the age of road segments within the derived road network. This approach uses the free and open-access Landsat archive [26] to produce yearly, gap-free, surface-reflectance image composites at a spatial resolution of 30 m [25], taken from all available Landsat TM (Thematic Mapper) and ETM+ (Enhanced Thematic Mapper Plus) images. The C2C approach also allows for the detection and characterization of forest disturbances [25]. For image compositing, a target date of August 1st $\pm$ 30 days is used, as this date coincides with the growing season for most of Canada's forested ecosystems. All atmospherically corrected Landsat images [27,28] acquired within this target time frame from 1984 to 2021 are used, along with pixel scoring functions, to create the best available pixel surface reflectance image composites. The pixel scoring functions rank and select the optimum annual pixel observation to be used in an image composite based on sensor type, date of acquisition, distance to cloud or cloud shadow, and atmospheric opacity [26]. The initial annual surface reflectance image composites generated suffer from data gaps due to varying data availability at different latitudes and persistent clouds in certain regions. To mitigate this issue, anomalous values are removed, and the data gaps are infilled using proxy values assigned through a spectral trend analysis [29]. This results in the generation of seamless Landsat surface-reflectance image composites from 1984 to 2021.

### 2.4. Existing Roads Database and Reference Field Data

The BC provincial road database provides contemporary information about the road network in BC. In its most recent update in 2018, 719,000 km of road was defined, with approximately 92% of them unpaved and used by the natural resource sector. The dataset is based on the Digital Road Atlas of British Columbia (https://www2.gov.bc.ca/gov/content/data/geographic-data-services/topographic-data/roads, last access: 12 June 2023), which combines map data from historical survey data and satellite imagery. However, both the spatial accuracy (i.e., the position of the road itself) and the accuracy of the presence or absence of roads (completeness and correctness) are unknown. The government of BC has acknowledged this lack of information on roads, suggesting that up to 150,000 km of resource roads are missing and the current data are inaccurate [12]. In the smaller study area presented in Figure 1, 6600 km of road are present in the database, which we use for training and validation. We exclude roads of the type "Trail" ("A pathway for pedestrians or non-motorized vehicles" according to the Dataset Attribute Dictionary) from our analyses, as they typically constitute non-drivable paths of subordinate relevance. Furthermore, they are often not detectable in the satellite imagery due to their limited width, resulting in closed forest canopies over the trails.

### 2.5. Methodology Overview

In Figure 2, we show a flowchart of the full pipeline presented in this paper. Note that for the determination of the road age and for the evaluation of the method, additional data are required.

We additionally provide the source code of our method on GitHub (Release 1.0.1, https://github.com/lwiniwar/roadCNN, last access: 29 February 2024) and have indexed it with Zenodo [30].

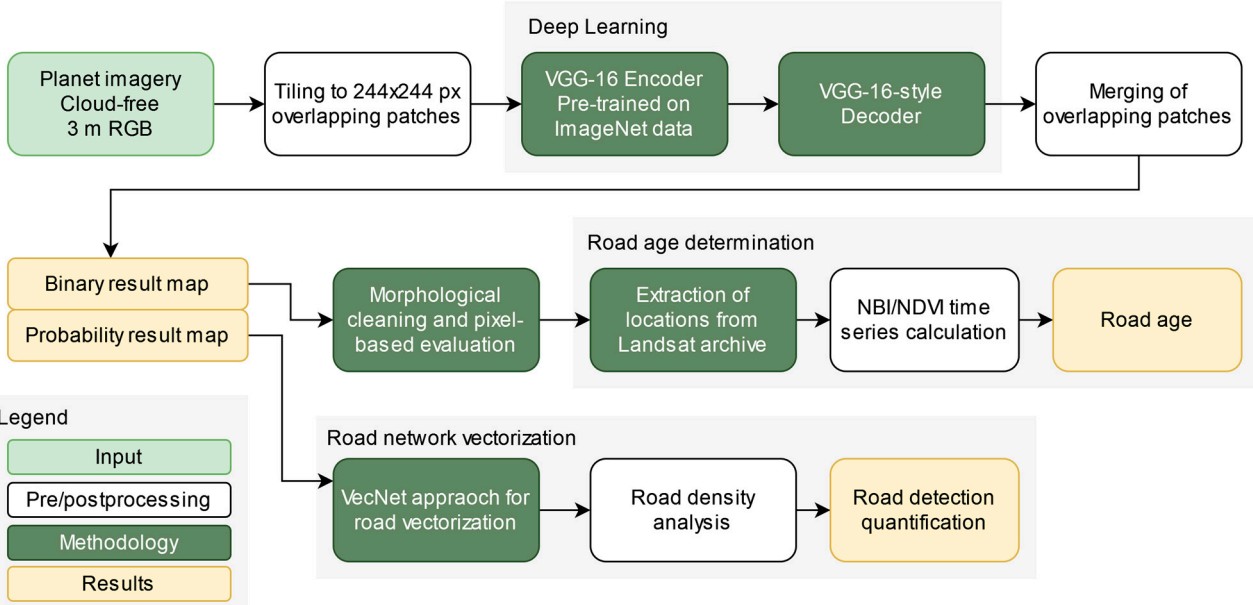

**Figure 2.** Flowchart of the method presented in this paper. Starting with Planet imagery (Section 2.2), we use a deep learning approach based on VGG-16 with pre-trained encoder weights to create a binary and a probability result map. From these maps, we determine road age and derive a vectorized road network.

### 2.6. CNN Road Classification

Recent advancements in deep learning and the development of Convolutional Neural Networks (CNNs) in computer science have led to a wide variety of applications utilizing remote sensing datasets. Developed from traditional neural networks, deep neural nets can outperform traditional machine learning methods used in environmental remote sensing [31]. A review of current advances in CNN road extraction confirms that the majority of studies to date have extracted urban road networks rather than rural road networks. Studies were predominately from China and the United States, with very few studies conducted in Europe, Canada, Asia, Africa, or South America. For new algorithm development, many studies used the Massachusetts Road Dataset [32] or the DeepGlobe Road Extraction Challenge 2018 Dataset [33] for both model training and validation. With respect to the accuracy of the existing approaches, most papers define their own evaluation metrics and datasets, making state-of-the-art accuracies difficult to derive.

A recent example is Microsoft's RoadDetections dataset, using a Residual U-Net neural network [15] on Bing Imagery. By visual inspection of the study area in the dataset provided by Microsoft (https://github.com/microsoft/RoadDetections, last access 19 February 2024), we found that many of the roads were missing, which we attribute to the network mainly having been trained on paved roads in urban environments.

We therefore employ a domain transfer technique, where a neural network with pre-trained weights is adopted. The use of this pre-trained network reduces the amount of training required as the solution space is pre-selected, limiting the amount of data required as well as increasing convergence speed. We use the SegNet approach [34], which consists of an encoder and a decoder network. For the encoder network, we apply weights from the pre-trained VGG-16 network [35], which used the ImageNet dataset [36]. As this network has been trained to identify 1000 different object classes from photographs (e.g., cats, different dog breeds, etc.), a domain transfer to satellite imagery and road detection is required. Such domain transfers have previously been shown to be efficient with satellite imagery (e.g., [37,38]). The decoder network is initialized with random weights.

The SegNet encoder consists of four convolutional blocks, each followed by a max-pooling layer. Similarly, the decoder is a sequence of upsampling layers, followed by

convolutional blocks. The output of the last layer is fed through a softmax layer to create pseudo-probabilities for the two classes (non-road and road). The dimensions and filter sizes for each of the layers are shown in Figure 3.

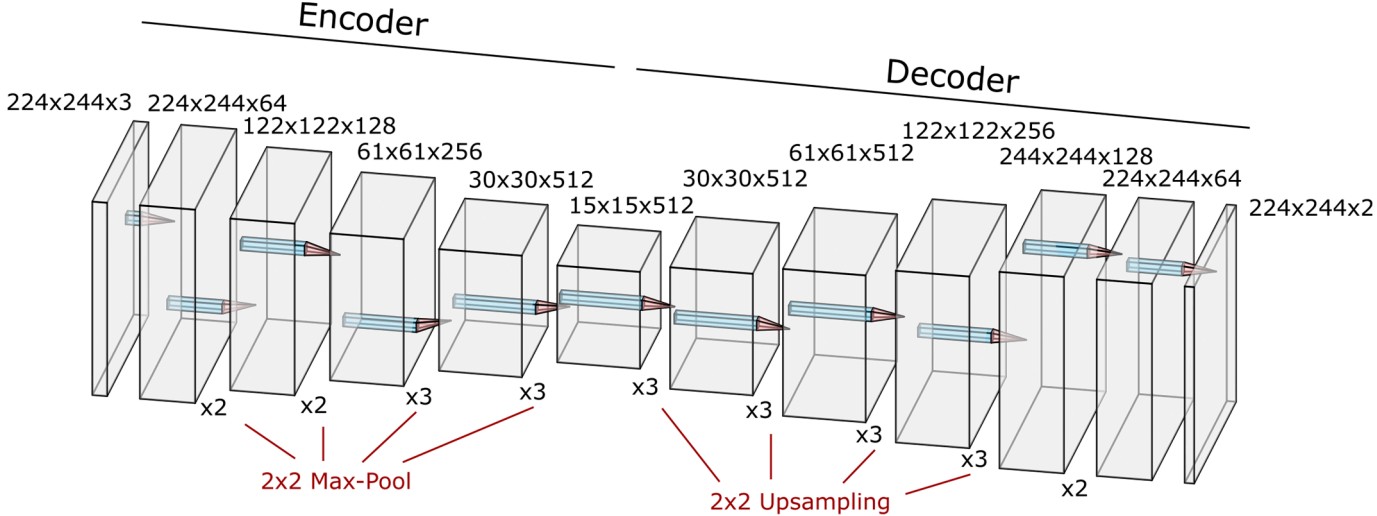

**Figure 3.** SegNet CNN architecture showing the encoder (on the left) and the decoder (on the right). The numbers above the blocks are the tensor dimensions (width × height × channels); the numbers below the blocks show how often each block is applied repeatedly.

To ensure that weight updates are focused on the decoder network, we set the learning rate of the encoder to ½ of the value for the decoder. The hyperparameters for the network are tuned manually, and the final values are presented in Table 1. We use weighted cross-entropy as a loss function to ensure that the network focuses on learning road pixels over non-road pixels, anticipating that false positives (road pixels where there are no roads) are less problematic than false negatives (missing road pixels) in the post-processing steps, especially with the vectorization step.

**Table 1.** Tuned hyperparameters used in the final network.

| Hyperparameter | Value | Extra Information |
|---|---|---|
| Learning rate | 0.0001 | Decreased after each epoch ($\gamma = 0.1$) The learning rate is halved for all weights of the encoder block. |
| Optimizer | Adam | Betas: $\beta_1 = 0.9, \beta_2 = 0.999$, Weight decay: $wc = 0.0005$ |
| Loss function | Weighted cross-entropy | |
| Weight balance | 0.95 (road), 0.05 (non-road) | |
| Data augmentation | 8-fold | Rotations by 90, 180, and 270 degrees, vertical and horizontal flips, and combinations (following [39,40]) |
| Number of epochs | 3 | 30,559 patches each |
| Batch size | 16 | Resulting in 163,408 batches |
| Input patch size | 244 × 244 pixels | |
| Patch overlap | 122 pixels | Each pixel is classified 4 times |
| Merge function | Piecewise linear function | The impact of a patch's pixel on the final result decreases with the distance from the center, but is capped at ¼ of the patch size (cf. Figure 4) |

### 2.7. Dataset Preparation

As the network architecture is limited to processing images of input size 244 × 244 pixels, patches of that size are created from the satellite imagery. To minimize edge effects, we create these patches with an overlap of 50%, i.e., 122 pixels. The patches are created on-the-fly from the larger raster datasets by the data loader.

For training and loss calculation, we use these patches, but when inferring the road network on the validation and test areas, we combine overlapping patches by a weighted average. Weights are calculated by a piecewise linear function (in x- and y-dimension) where weights are decreasing from the center, and the maximum is capped at ¼ of the patch width/height (Figure 4). The reasoning here is that the CNN is able to make better predictions in the image center as the neighborhood context is clearer. Thus, each pixel's road probability is the result of four passes through the network. To obtain a final binary map for pixel-wise comparison, simple thresholding with a cut-off value of 0.5 is used.

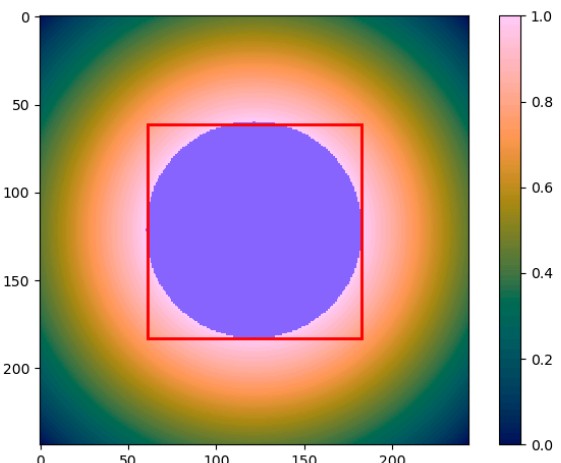

**Figure 4.** Weights within each patch used for merging. The red square indicates a distance of ¼ of the patch size from the border (61 px), and the blue circle shows the central area where all weights are equal to 1. Consequently, the pixels in the corner of each patch have minimal influence on the final result as their weights are close to zero.

The binary reference dataset (the labels) for both training and evaluation (Section 2.8) is created by using the BC provincial road database (Section 2.4). The road data are vectorized using GDAL [41], taking care to align the resulting raster with the RGB imagery. Pixels that intersect any of the filtered roads are set to value "1", while the background is set to value "0". To emulate road width representation in the binary training data, we morphologically dilate the data. We obtain road widths of 2 px or 6 m, which are typical values for the roads in our study area.

In the training process, we limit input patches to ones that contain at least 10 pixels classified as roads. This data pre-selection speeds up training significantly, as patches with no roads present provide little information on how to detect roads. For validation and testing, we run the network on full data, as no a priori information on road presence is assumed.

### 2.8. Post-Processing and Evaluation

As the training, test, and validation road network data are not perfectly aligned to the satellite imagery, we anticipate that initial performance scores are quite low, as a large number of false positives and false negatives occur at the edges of the detected roads. More specifically, with a 3 m pixel (px) size, the Planet imagery is likely to show changes in radiance within 1–3 px. The equivalent would be a 9 m side road, which is unlikely to occur. As we rasterize the reference dataset, roads in this dataset are 3 m (1 px) wide. Therefore,

lots of false positives are detected along the road if the classification output is more than 1 px wide (Figure 5).

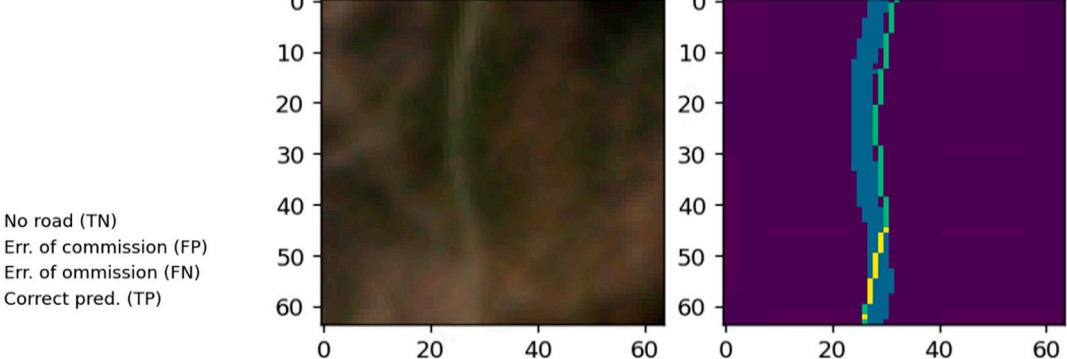

**Figure 5. Left**: RGB imagery with a road visible. **Right**: Classification result (true positives and false positives) of the road. It can clearly be seen that the detected road follows the reference road at this position, yet only a small part of the road is considered correctly predicted ("True Positive", yellow).

As we are principally concerned with the detection of a road, and less with its exact position within a 9 m buffer, we employ morphological operations to disregard these misclassifications. To that end, we first buffer the reference roads as well as the predicted roads by *n* pixels using morphological dilation. We then select all the pixels in these buffered maps that are true positives, i.e., correctly predicted roads. From the map of true positives, we again apply a buffer, this time of 2*n* pixels. The buffer size *n* is determined in accordance with the expected alignment error, maximum road width, and pixel size. We investigate different values for *n* and observe a convergence limit for the evaluation scores.

The resulting buffered map is used as a mask. In this area, false negatives and false positives are disregarded as being the result of alignment and road width mismatches. Subsequently, the number of true positives (*TP*, correctly identified road pixels), true negatives (*TN*, correctly identified non-road pixels), false positives (*FP*, non-road pixels incorrectly identified as road), and false negatives (*FN*, road pixels incorrectly identified as non-road) are evaluated, and metrics of accuracy, precision, recall, and F1-Score are calculated according to Equations (1)–(4).

$$Accuracy = (TP + TN)/(TP + TN + FP + FN) \tag{1}$$

$$Precision = TP/(TP + FP) \tag{2}$$

$$Recall = TP/(TP + FN) \tag{3}$$

$$F1 = 2 \cdot (Precision \cdot Recall)/(Precision + Recall) \tag{4}$$

To further understand the impact of different topographic factors on the classification quality, we investigate the four metrics above stratified by different slope angle ranges, slope aspects, land cover classes, and Normalized Difference Vegetation Index (NDVI) values. The slope angle and aspect are calculated from a 30 m DEM [42,43], which is subsequently resampled to fit the 3 m cell size of the RGB data using nearest-neighbor interpolation. Similarly, the 30 m land cover map [25] is resampled using nearest-neighbor interpolation. The NDVI is calculated from the input RGBI data and therefore does not need to be resampled.

### 2.9. Vectorization

For certain applications, such as navigation, a topologically intact road network is essential. To automatically derive such a network from satellite imagery, we use the method outlined by [21] and implemented in an R package [44]. This method operates on the probability raster typically generated by machine learning techniques.

The vectorization employs a "driving" approach. It begins with seeds of roads, manually created or automatically detected (manually created in our study), and digitally "drives" along the probability raster by tracking high-probability pixels, utilizing the concept of friction of distance [45–48]. At each step, the algorithm follows a short road section with a limited line of sight to anticipate obstacles and record intersections. The algorithm is continuing until there are no intersections or valid pixels left to drive on. This approach inherently cannot reach isolated roads disconnected from the initial seeds; thus, we assume that road networks are connected, without any disconnected 'islands' within the analysis area. Additional constraints, such as minimum probability, maximum line of sight, and viewing angle, can be adjusted.

Due to single patches not being able to predict roads correctly, gaps in the raster probability map appear. We tuned the parameters of the vectorization algorithm such that it was able to only leap over small gaps in the road network.

On the large area of interest, we first merged the predictions for the individual over-passes by averaging the probabilities in overlapping areas. Subsequently, we vectorized roads by using the full existing road network layer as seeds for the algorithm. This follows the application of a road update focusing on additionally built roads, as we assume that even discontinued roads have a long-lasting spectral footprint. In the results, we therefore only report on additional roads being detected, with the caveat that these roads may also be false positives (as described above).

*2.10. Road Age Determination*

The annual Landsat image composites were clipped to the derived CNN road network of the small test tile (see Figure 1). We derived spectral profiles for each pixel over time by calculating their associated NDVI and Normalized Burn Ratio (NBR) values. NDVI has been widely utilized to track changes over a time series of satellite imagery as it is a reliable measure of vegetation productivity and health [49–51]. The time series of NBR pixels has been demonstrated to be a sensitive and consistent method for the retrieval of disturbance events in forest environments [52]. The construction of a road through a forested environment is associated with a sudden shift in the spectral properties, resulting in NDVI and NBR values decreasing. To detect this change, a pixel-wise change detection algorithm was developed and applied over the time series (Figure 6). When there is no drop in spectral values, it is assumed that the road is either too small for a noticeable drop in indices to occur, that the road was built in or prior to 1984 (the start of the time series), or that the detection does not represent a road (i.e., a false positive).

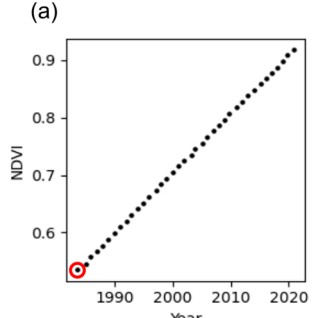 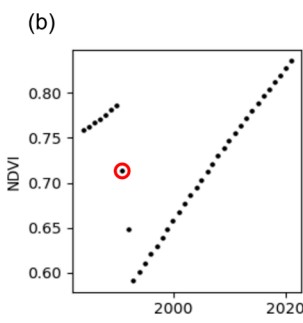 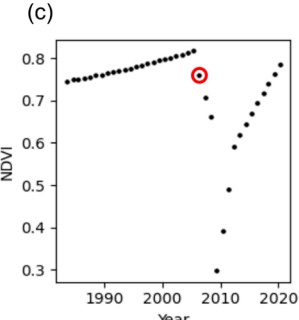 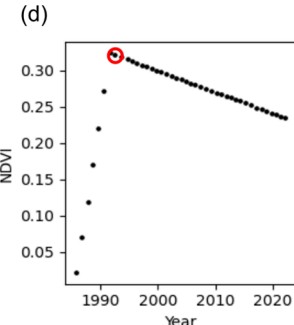

**Figure 6.** Spectral profiles of selected pixels through time. In Examples (**b**,**c**), a clear point of disturbance is identified, and can be attributed to the construction of a road. Example (**a**) shows a road that was constructed before 1984, and Example (**d**) shows a steady decrease in NDVI from the year 1990 onward. The red circles indicate the year of the detected change.

## 3. Results

Selected areas of the RGB test data are shown in Figures 7–9, along with the ground-truth and the predicted road probability. While most roads, both paved and unpaved, are identified reasonably well from the imagery (Figure 7), the CNN estimate gets poorer when roads disappear beneath the canopy. Roads located in active mining areas (Figure 8) and/or recent clear-cuts (Figure 9) are typically not detected. Especially in these areas, the quality of the reference data is also poorer, as roads often change over short periods, and the road database is not updated frequently. Furthermore, the local contrast between the road and the disturbed forest is less than between the road and the undisturbed forest. Another issue arises with seismic lines used for geophysical exploration, which result in similar patterns in satellite imagery as roads (i.e., linear features of disturbance). Therefore, they are typically delineated as roads (cf. Figure 7).

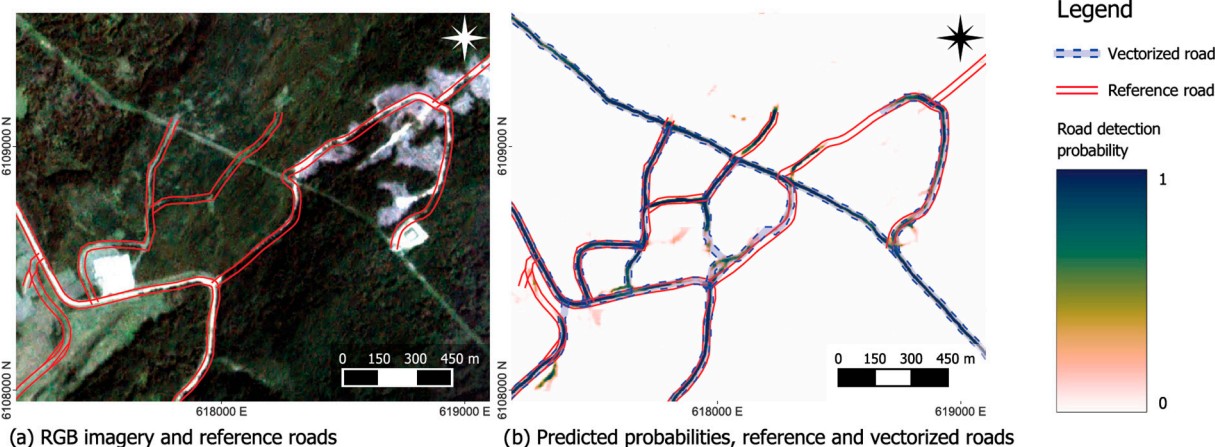

**Figure 7.** Detailed vectorization results showing a seismic line incorrectly classified as a road, with good classifications otherwise. (**a**) RGB imagery and reference road network; (**b**) predicted probabilities, reference road network, and predicted road network after vectorization.

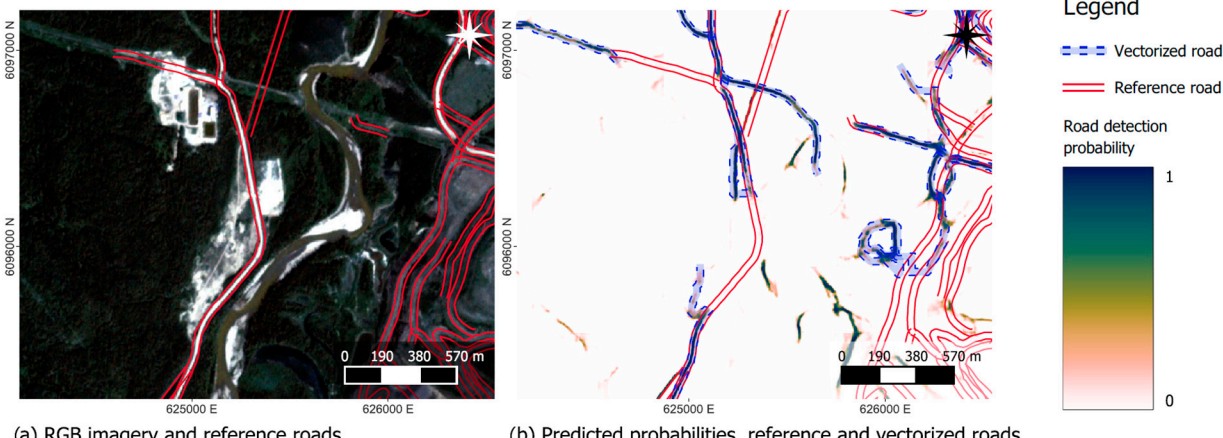

**Figure 8.** Detailed vectorization results showing a mining area (eastern part) and barren ground around infrastructure (center, along the road), where roads are generally missed. (**a**) RGB imagery and reference road network; (**b**) predicted probabilities, reference road network, and predicted road network after vectorization.

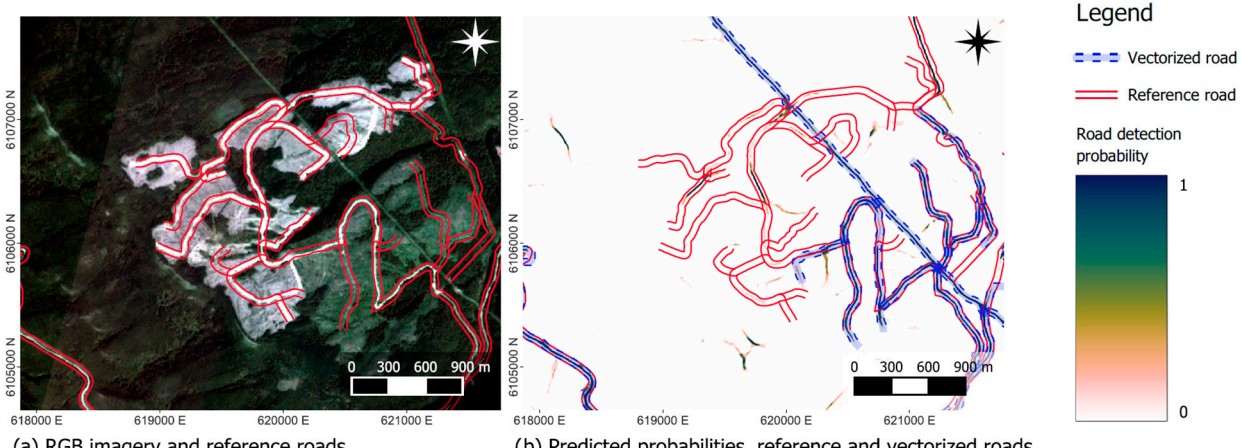

(a) RGB imagery and reference roads

(b) Predicted probabilities, reference and vectorized roads

**Figure 9.** Detailed vectorization results showing a freshly clear-cut area (white), where roads are generally missed, and an older clear-cut area (light green), where the detection works well. (**a**) RGB imagery and reference road network; (**b**) predicted probabilities, reference road network, and predicted road network after vectorization.

Spatial offsets between the reference vector data and the imagery can be seen, e.g., in Figure 5. These offsets cause a large proportion of the false positives and false negatives, as shown in Table 2. Consequently, the measures of precision, recall, and F1 are influenced by these offsets. Using morphological dilation operations (as presented in Section 2.8), we acknowledge this spatial offset and can obtain improved estimates of the performance scores. By further increasing the allowed offset, we also incorporate parts of the mining- and freshly clear-cut areas (cf. Figures 8 and 9), where false positives and false negatives are ignored. Figure 10 shows the progression of the performance metrics for increasing buffer sizes, which are calculated from the number of true positives, true negatives, false positives, and false negatives of the test tile, as shown in Table 2. While recall values (how many of the roads are detected) are very low for small dilation values, they increase quickly and are already >50% after a 2-pixel buffer (corresponding to a 6 m offset). The precision (how many of the detected pixels are roads) has a slower increase, indicating that the number of false positives is less of a result of spatial misalignment. Still, precision converges to a value of about 95%, while recall reaches 99% at a 15-pixel (45 m) buffer around the roads.

**Table 2.** Prediction quality for different buffer sizes, measured in pixels for the test tile. The large number of false positives for small buffer sizes mostly corresponds to spatial alignment problems in the datasets.

| Buffer Size | True Positives | True Negatives | False Positives | False Negatives |
|---|---|---|---|---|
| 0 px (output) | 172,666 | 111,843,439 | 2,012,961 | 360,545 |
| 1 px (3 m) | 584,586 | **112,262,164** | 1,232,338 | 310,523 |
| 5 px (15 m) | 2,489,847 | 110,880,575 | 779,587 | 239,602 |
| 15 px (45 m) | **7,859,615** | 105,864,302 | **532,243** | **133,451** |

In Table 3, we show how the CNN road extraction performs on different terrain features, i.e., slope, aspect, elevation, as well as crown coverage (indicated by the NDVI) and land cover classes. When stratified by NDVI (indicative of canopy cover), higher accuracy values correspond to high NDVI values. However, precision in particular decreases with high NDVI, which corresponds to incorrectly labeled roads under dense vegetation. This holds mostly true when a 15 m buffer is considered. The recall value is highest for medium NDVI values between 0.5 and 0.75.

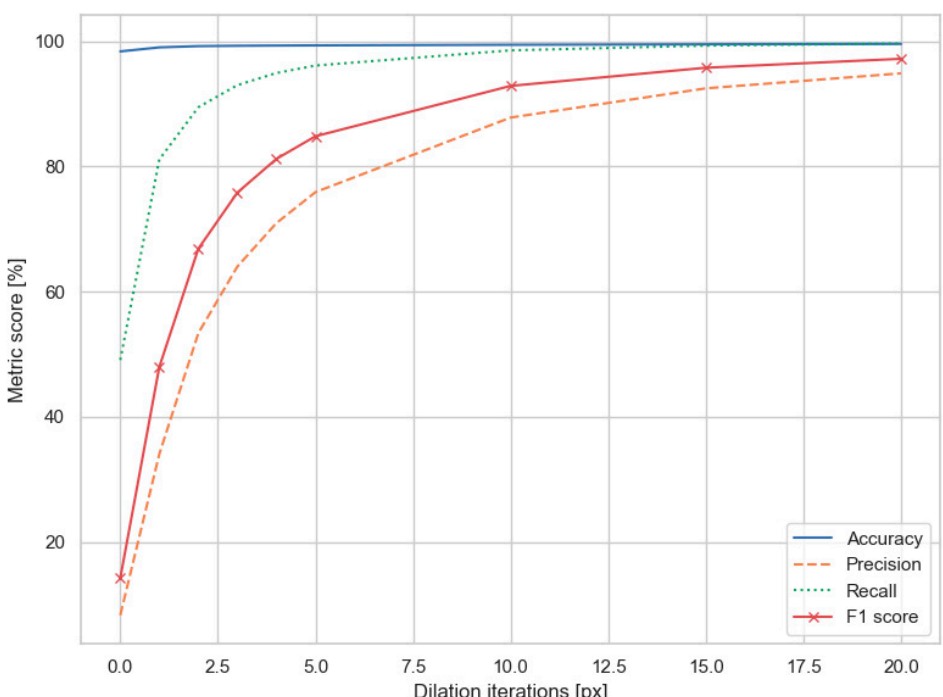

**Figure 10.** Accuracy, precision, recall, and F1-Score for the binary classification, shown for different buffer sizes (dilation iterations). One pixel corresponds to 3 m of allowed offset.

**Table 3.** Performance values stratified by different land cover types, vegetation coverage, slope, and aspect. Values in parentheses are after allowing for a 5 px (15 m) dilation. N/A values correspond to no true positives in the respective strata.

|  | Accuracy [%] | Precision [%] | Recall [%] | F1 [%] |
|---|---|---|---|---|
| NDVI < 0 | N/A (98.6) | N/A (60.0) | N/A (100.0) | N/A (75.0) |
| 0 ≤ NDVI < 0.5 | 95.0 (97.8) | 13.6 (68.1) | 50.0 (88.2) | 21.4 (76.8) |
| 0.5 ≤ NDVI < 0.75 | 91.7 (97.6) | 11.5 (81.2) | 61.7 (96.4) | 19.4 (88.1) |
| 0.75 ≤ NDVI < 1 | 98.9 (99.5) | 6.1 (74.6) | 40.9 (96.6) | 10.6 (84.2) |
| Slope < 5 deg | 98.1 (99.3) | 9.2 (79.9) | 51.3 (96.4) | 15.6 (87.4) |
| 5 ≤ Slope < 10 deg | 97.9 (99.2) | 8.9 (79.2) | 50.4 (96.4) | 15.2 (86.9) |
| 10 ≤ Slope < 20 deg | 98.5 (99.3) | 7.8 (73.5) | 46.9 (95.8) | 13.4 (83.2) |
| 20 ≤ Slope < 30 deg | 99.1 (99.4) | 5.0 (54.8) | 38.9 (93.9) | 8.8 (69.2) |
| 30 ≤ Slope | 99.3 (99.4) | 1.7 (25.6) | 24.9 (90.2) | 3.2 (39.9) |
| Aspect: North | 98.2 (99.3) | 8.5 (75.9) | 47.6 (95.7) | 14.4 (84.7) |
| Aspect: East | 98.3 (99.3) | 8.7 (77.2) | 51.6 (96.5) | 15.0 (85.8) |
| Aspect: South | 98.4 (99.4) | 8.5 (77.0) | 51.4 (96.5) | 14.6 (85.6) |
| Aspect: West | 98.5 (99.4) | 7.9 (74.4) | 48.3 (96.1) | 13.6 (83.9) |
| LC: 20 (Water) | 99.5 (99.6) | 1.0 (23.1) | 40.6 (100.0) | 1.9 (37.5) |
| LC: 31 (Snow/Ice) | 98.2 (98.9) | 5.1 (55.4) | 54.5 (97.5) | 9.3 (70.7) |
| LC: 32 (Rock/Rubble) | 98.0 (98.1) | 0.0 (4.0) | 1.0 (64.9) | 0.1 (7.5) |
| LC: 33 (Exposed/Barren Land) | 96.0 (98.0) | 6.6 (67.0) | 28.2 (88.0) | 10.8 (76.1) |
| LC: 50 (Shrubs) | 94.4 (98.5) | 10.2 (86.3) | 55.4 (97.4) | 17.3 (91.5) |
| LC: 80 (Wetland) | 97.1 (99.0) | 9.8 (80.3) | 66.9 (98.5) | 17.1 (88.5) |
| LC: 81 (Wetland-treed) | 98.2 (98.8) | 4.3 (53.9) | 60.1 (99.2) | 8.0 (69.8) |
| LC: 100 (Herbs) | 92.0 (98.6) | 12.0 (91.8) | 56.7 (97.4) | 19.8 (94.5) |
| LC: 210 (Coniferous) | 99.2 (99.6) | 7.1 (68.2) | 51.4 (96.7) | 12.5 (80.0) |
| LC: 220 (Broadleaf) | 98.1 (99.1) | 6.9 (70.3) | 45.7 (95.9) | 12.0 (81.1) |
| LC: 230 (Mixed wood) | N/A (99.9) | N/A (60.0) | N/A (75.0) | N/A (66.7) |

Across different slope values, there is little variation in the accuracy, especially when considering the 15 m buffer. However, both with and without buffer, higher slope values correspond to a decrease in precision and recall, in turn resulting in lower values of F1 score. This is one of the strongest relationships observed among the stratification. In contrast, road aspect has a minimal influence on the road detection capabilities, as values are almost constant across the different cardinal directions indicating the detection is not influenced by, for example, shaded, north-facing hills.

The highest detection accuracy is achieved in the coniferous land cover class, which is the most prominent non-road class. For land cover, recall scores are especially interesting, as they explain how likely a road is to be detected in a specific land cover. Overall, the lowest performance is achieved on rock/rubble and exposed/barren land, which correspond to mining sites and clear cuts. In contrast, precision scores explain how many of the pixels detected as roads are actually roads in the reference data. Here, the lowest values are achieved for the land classes of rock/rubble, water, and snow/ice. The latter two classes do not appear often in the dataset, but are spectrally similar to roads, which may explain the confusion. The highest precision is achieved for herbs, shrubs, and wetland classes, where roads are typically not overgrown by tall vegetation, resulting in a prominent spectral footprint.

In the vectorization step, 870.5 km of roads are vectorized from 75 initial, automatically derived seeds for the test area (cf. Figure 1). The resulting road density map, derived from the vectorized road data, is presented in Figure 11. For validation purposes, the road density map of the reference dataset is displayed next to it. Furthermore, we calculate the difference in road densities (Figure 12). In contrast to the absolute values provided by the classification metrics, these maps show the overall spatial prediction quality.

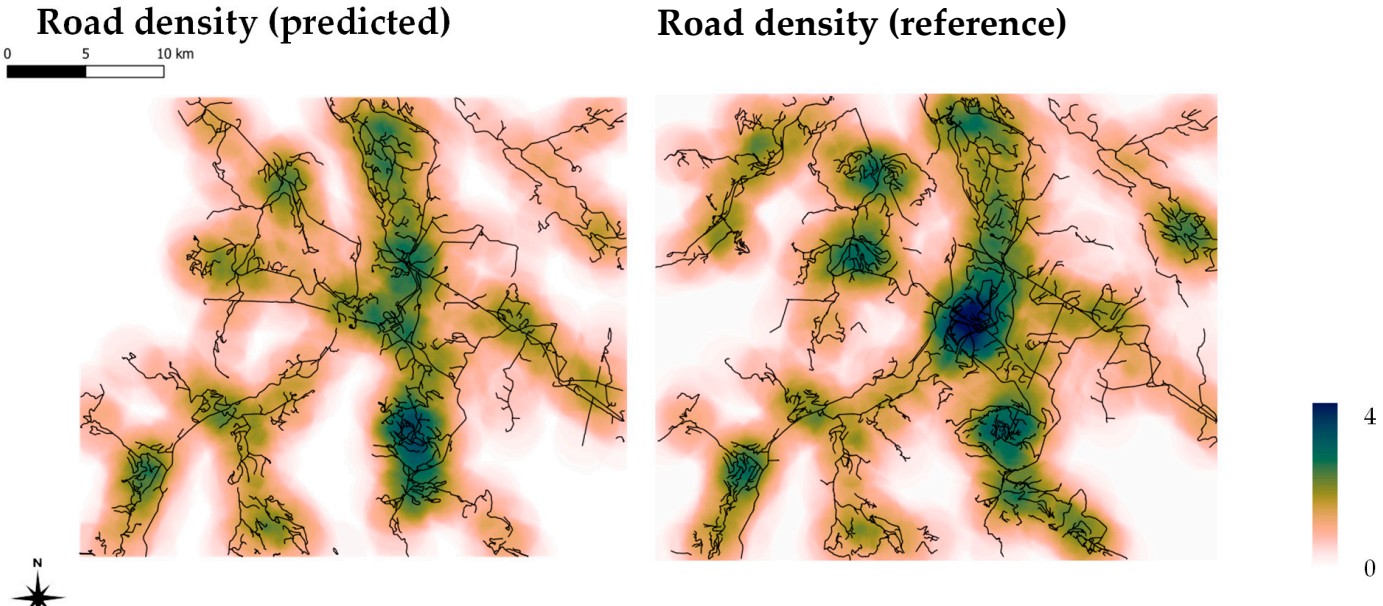

**Figure 11.** Predicted (**left**) and reference (**right**) road densities. The road densities were calculated on a 250 m grid using a search radius (range) of 2 km.

In Figure 12, differences in road density between the reference data and the result after the vectorization step are shown. In general, difference values are low (within $+/-1$ km/km²) over the investigated area. The two main differences are missing roads in the North-West, where a single road link is missing and the vectorization is interrupted, and in the center, where a mining operation has many roads in the reference data that are not present in the predicted dataset due to their radiometric footprint.

On the larger area of interest, segments were derived from the reference road layer for multiple overlapping satellite data strips. After merging these data, a total length of 14,358 km of roads was digitized automatically from the imagery. To investigate the difference to the reference road dataset, a buffer of 15 m around reference roads was established, and predicted road lengths were summed up when they intersected the larger area of interest but not the buffer (i.e., excluding roads close to the ones present in the reference data). This resulted in 5774 km (40.2% of all detected roads) of roads that were detected in addition to the reference layer. In contrast, the reference network in the larger area of interest consisted of a total of 19,518 km, of which the CNN approach successfully detected only 8649 km (44.3%).

When applying the change detection algorithm over the NDVI time series, it was found that 51% of roads were built post-1985, with 30% of roads being built prior to 1984 and 19% around 1984/5. The output and distribution of road age when applying the change detection method over the NBR and NDVI time series are illustrated in Figures 13 and 14, respectively. When applying the change detection algorithm over the NBR time series, it was found that 49.5% of roads were built post-1985, with 26.5% of roads being built prior to 1984 and 25% in 1984/5.

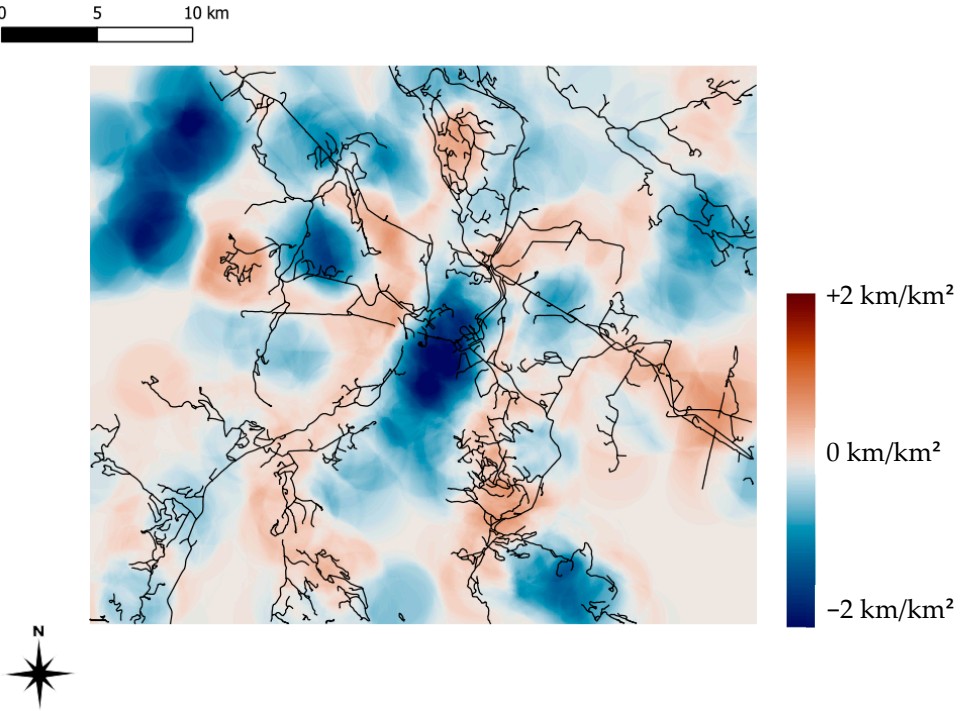

**Figure 12.** Differences in derived road densities. Negative values (blue) correspond to roads present in the reference dataset, but not in the predicted data. Positive values correspond to more detected roads than what was present in the reference dataset. The displayed road network is the predicted vectorized data.

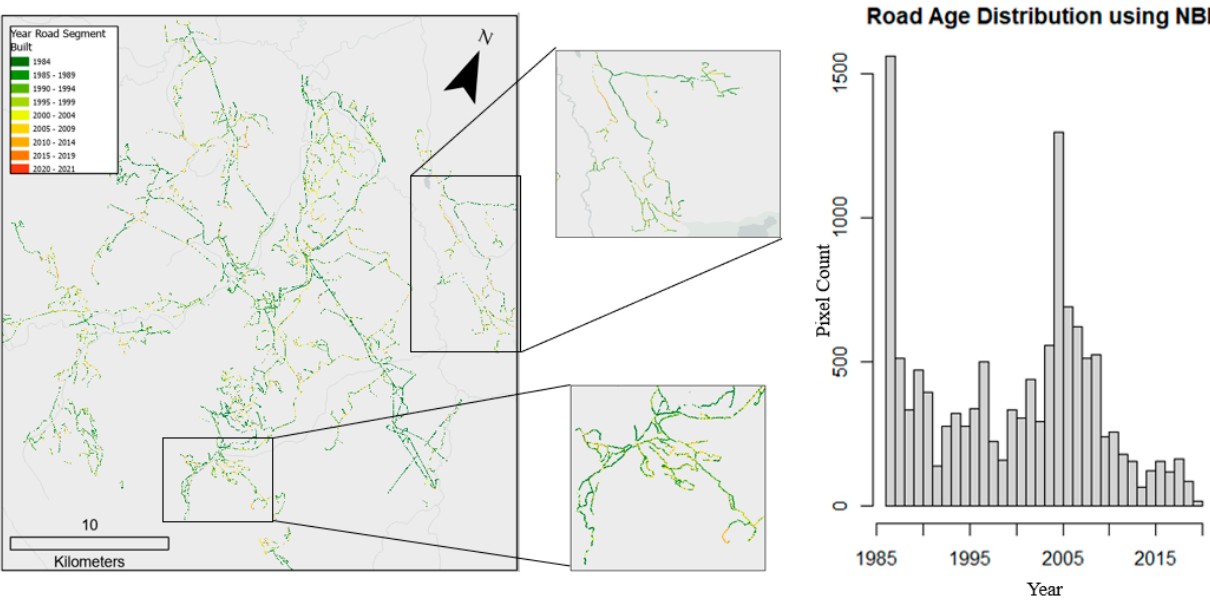

**Figure 13.** Output of the change detection algorithm applied over NBR time series, with supporting insets. Road age is grouped into 5-year time steps; 49.5% of roads were built post-1985; the histogram displays the distribution of roads built in this period.

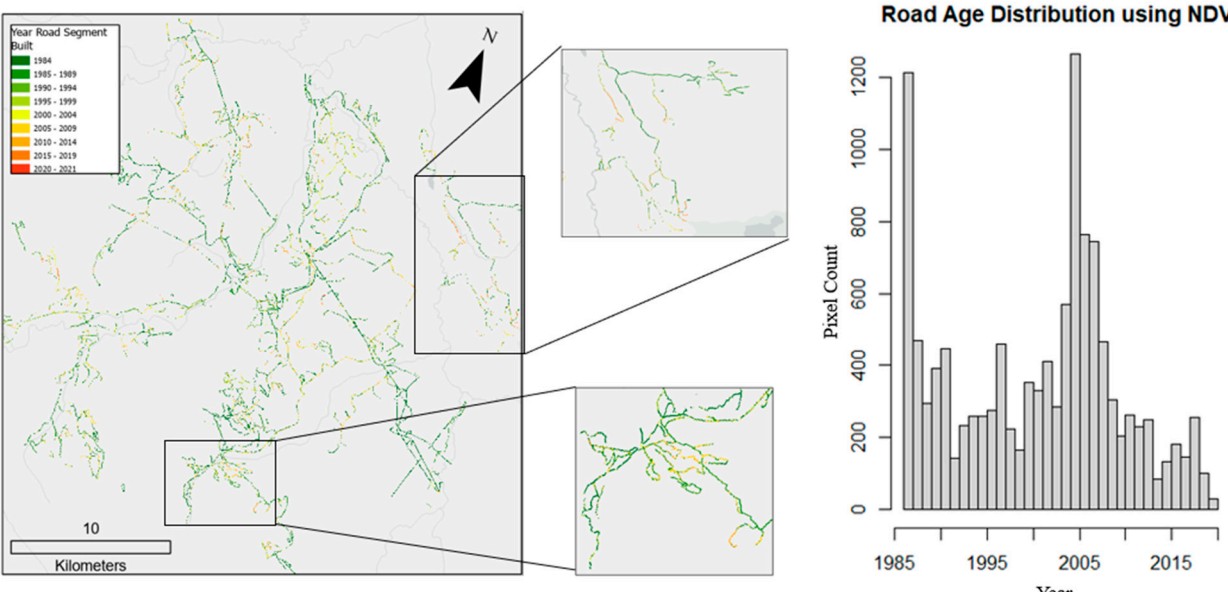

**Figure 14.** Output of the change detection algorithm applied over the NDVI time series, with supporting insets. Road age is grouped into 5-year time steps; 51.0% of roads were built post-1985; the histogram.

## 4. Discussion

The CNN-based road extraction approach developed in this study and trained on Planetscope imagery, achieved reasonably high performance on both the paved and unpaved rural roads typical of the boreal regions of Canada. Recall, precision, and F1 quality metrics were comparable to studies using higher-resolution imagery, such as Rapideye, to extract paved roads across much smaller areas using deep learning (e.g., [53,54]) and other approaches (e.g., [55,56]). A key benefit of using data from CubeSats, such as PlanetScope, is its high spatial and temporal resolution. While PlanetScope imagery is typically available at daily or near-daily timesteps, clouds and cloud shadows increase the revisit period,

which can, in cloudy areas, exceed a number of weeks [57]. In most cases, this delayed temporal resolution is unlikely to cause issues, as the updating of road information is unlikely to occur more often than a few times per year. PlanetScope data products use anomalous brightness values to detect clouds. However, recently constructed roads with little remaining vegetation will also have anomalous brightness values and may be misclassified as clouds, thereby reducing the number of valid pixels from which to detect a change. With improvements in the UDM2 product and the addition of new spectral bands to PlanetScope satellites, this misclassification of clouds may be reduced [57]. In addition, while users can download PlanetScope scenes with a daily temporal revisit, Planet now offers image composites at coarser temporal resolutions (e.g., biweekly or monthly), which typically have lower cloud cover than that of the average scene over a given area.

A key benefit of the approach used is the use of a pre-existing training set with which to build the initial CNN model. The use of deep learning algorithms such as CNN has expanded greatly within the past 5 years, and advances in computing power and the ability to apply deep learning algorithms in a spatial context allow users previously unable to utilize the power of CNN to now be able to do so. Roads are difficult objects to classify due to their spectral similarity to other high-albedo objects such as bare soil and rock (corresponding to the areas where we achieved the lowest classification performance) and their morphological similarity to other linear features such as rivers and pipelines, making a CNN approach logical to apply.

Another key outcome of the methods developed in this paper is the development of a topologically connected road network, rather than separate, discrete road segments. Even after post-processing, a pixel-based road extraction approach can be difficult to convert into a connected vectorized road network for navigation [54]. The approach, developed by Roussel et al. [21] and applied here, allows the detected road segments to be connected in a vector dataset to ensure a topologically correct road network, which is required for most applications such as road planning and conservation and needed in order to ingest this road information into existing vector databases. Note that we do not detect any over- or underpasses as disconnected parts, which would need to be manually corrected in the road network. However, these rarely occur in rural forest road settings.

As shown in Figures 7–9, the CNN sometimes fails to detect parts of the road, resulting in gaps in the binary road layer. One of the advantages of the vectorization method we use is that small gaps—typically up to 50 m—can be bridged, as long as the detected probability does not drop too low and the gap does not occur at a tight turn [21]. Finally, if a full network is required, manual correction and editing can be carried out using standard GIS software.

While the geographic area chosen for our study did not cause any issues with permanent cloud cover, we acknowledge that this may be different for other areas and temporal periods. In the case of the Canadian boreal, forest fires have been increasing in magnitude and duration in recent years, which may cause atmospheric distortions. A successful detection of the roads using RGB satellite data under these conditions would not be possible.

Lastly, our analysis indicates how long-term archival satellite data can be used to determine the age of forest roads in northern British Columbia using time series of spectral indices and annual Landsat best-available pixel image composites. The road age results produced from the NDVI and NBR time series are comparable, with minor variations from year to year. Both methods, however, result in road segments along a road fluctuating in age, despite the segment likely having been built in a single year. This fluctuation in age is in part attributed to road segments within Landsat pixels being too small or affected by spectral noises from background land cover, resulting in no changes being detected. To account for this, future work will incorporate a minimum change detection threshold.

The histogram of road ages shows two peaks, one around 1984/5 and another around 2005. The former peak is related to the change detection algorithm, where a decreasing NDVI or NBR in the first years of observation will result in the first year being detected as the year of change (cf. Figure 5). The second peak in 2005 is not explainable by the

algorithm alone, and further investigations requiring additional reference data (e.g., GPS tracks of logging trucks) should be carried out in the future.

Up-to-date information on the rural road network is critical for the management of threatened and endangered species and is becoming a critical piece of information to aid in conservation activities. In British Columbia, with increasing levels of disturbance due to insect infestation, harvest, and fire, as well as ongoing anthropogenic disturbances associated with exploration and mining, the density of roads across the region remains of paramount concern. Quantifying the density of the road network within individual caribou herd ranges is one metric that can provide some context for how much activity may be occurring within each of these caribou herd ranges. This idea is similar to that undertaken with the management of the grizzly bear in Alberta, where the Albertan provincial government recommended road densities below certain thresholds to encourage the recovery of the species [58]. While these thresholds are important, it is therefore critical that spatial coverages showing the extent of the road coverage are accurate and complete, which is often not the case in remote areas. The technique presented here—detecting roads from high-spatial-resolution Planet imagery and then providing an indication as to their age from historic Landsat imagery—provides approaches to updating and partially validating existing road networks. In addition, the ability to vectorize these predictions in order to produce a linked road network, while in its infancy, is the critical final step in ensuring these layers can then effectively be used for management purposes.

The large disagreement between roads in the reference layer provided by the government and the extraction results, with only 44.3% of the reference roads detected and 40.2% of the detected roads not being present in the reference dataset, underlines the uncertainty in governmental datasets as well as in ones derived from satellite imagery. In parts, this is due to temporal changes (road construction and road decommissioning), but also due to the limited visibility of narrow roads even in high-resolution imagery and after excluding roads of type "Trail". The presented remote sensing-based CNN method should, therefore, not replace, but rather complement existing datasets and point to locations where further investigations can be carried out to improve the quality of the dataset. The dynamic nature of forest roads, coupled with the remoteness of the investigated area, makes fully automatic validation impossible.

Looking to the future, e.g., for the management potential of these types of technologies, one possible strategy would be to integrate GNSS data acquired from vehicle loggers. This would map the use of the landscape by vehicles and could be combined with geospatial coverage from satellite data, as demonstrated here. A result could be a database of both accurate road location information and road use. Information on the speed and diurnal use of roads could also be extracted from these databases to provide a comprehensive analysis of the road infrastructure in an area. While these types of databases are starting to be developed within the forest industry using GNSS receivers for harvesting operations, their broader use in a conservation setting has not yet been fully examined and may provide additional insights into road networks, particularly for endangered species.

## 5. Conclusions

High-resolution satellite imagery is readily available for large parts of the world and can be utilized in machine learning approaches to extract valuable geographic information. In our use case, we used a pre-trained Convolutional Neural Network (SegNet architecture) to extract the probability of forest road occurrence at the pixel level. The resulting probability map was then fed to a vectorization method, resulting in a topologically valid road network layer. Such a layer can be used, e.g., for routing tasks or for road density estimations.

While overall satisfactory and use-case-appropriate results were obtained, the presented method has certain limitations. First, the roads need to be visible in the satellite imagery to be detected, and canopy cover over road segments can hinder accurate detection. A clear contrast of the road surface to the surrounding areas must occur, which may prove

problematic in clear-cuts and similar settings. In road detection, the CNN sometimes failed to detect linear features well. Larger patches, i.e., larger fields of view of the neural network, could help to resolve ambiguities regarding these linear structures. With different network architectures having a larger field of view or context with which to learn, better results may be obtained.

However, existing road network data, while available, is often outdated at the time of publication, as data acquisition, mapping, and cleaning are labor-expensive tasks. Our approach shows how large areas (23,500 km$^2$) can be automatically processed in short times, allowing for frequent and inexpensive updates of road network information.

Additionally, we showed how the precise location data of the extracted roads can be combined with lower-resolution historical satellite imagery, allowing the derivation of road age. As a result, the development of road access in these forested areas can be mapped when Landsat imagery is cloud-free and available. However, this would constrain the analysis to post-1984. This allows for additional analyses, for example, on the cumulative effects of the developing road network on wildlife such as caribou.

**Author Contributions:** Conceptualization, N.C.C. and A.T.F.; methodology, L.W., N.C.C., J.-R.R. and A.B.; software, L.W., J.-R.R. and A.B.; validation, L.W.; formal analysis, L.W. and N.C.C.; investigation, N.C.C. and A.T.F.; resources, N.C.C.; data curation, D.Q.R.Z. and L.W.; writing—original draft preparation, L.W. and N.C.C.; writing—review and editing, L.W., N.C.C., A.B., J.-R.R., D.Q.R.Z., C.T.L. and A.T.F.; visualization, L.W. and A.B.; supervision, N.C.C.; project administration, N.C.C., C.T.L. and A.T.F.; funding acquisition, N.C.C., C.T.L. and A.T.F. All authors have read and agreed to the published version of the manuscript.

**Funding:** This research was funded by Environment and Climate Change Canada.

**Data Availability Statement:** The data can be obtained from Planet with an appropriate license. The source code of the developed method is available on GitHub (https://github.com/lwiniwar/roadCNN/, last access: 29 February 2024) and indexed with Zenodo [30].

**Acknowledgments:** We thank Chris Mulverhill for support in downloading and preprocessing the Planet CubeSat imagery. We thank J. Hughes and R. Steenweg for thoughtful comments on a previous version of this manuscript. We further thank the four anonymous reviewers for their helpful remarks.

**Conflicts of Interest:** The authors declare no conflicts of interest.

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
