# Peer review of "Extraction of Forest Road Information from CubeSat Imagery Using Convolutional Neural Networks"

_remotesensing, doi:10.3390/rs16061083_

Round 1

Reviewer 1 Report

Comments and Suggestions for Authors

Line 81: The acronym CNN is used for the first time here but without being expanded. Acronyms should be expanded at first use, and the acronym placed in brackets.

Line 84: The acronym NDVI is used for the first time here but without being expanded. Acronyms should be expanded at first use, and the acronym placed in brackets.

Line 139: The paper mentions that Landsat image composites were utilised, however there is no mention of whether it was Landsat 8, 7, or older satellites. Given that each Landsat satellite is very different from previous one, it would be helpful to mention which one was the source of the image composites.

Line 317: The acronym NBR is used for the first time here but without being expanded. Acronyms should be expanded at first use, and the acronym placed in brackets.

Line 324: This line says the time series is shown in Figure 4 but I think it is shown in Figure 5.

Line 446-452: This is a minor comment and can be ignored. The paragraph describes an observation made regarding when the detected roads were built. It would be great if the authors could add one or two sentences stated what the observations suggest about the roads or the machine learning model.

Line 485: The paper suggests that the model could be used for development of a fully connected road network. 

The word ‘fully’ is an over-stretch, given that the road extraction assumes that all road segments topologically intersect. Roads that pass under or over other roads could be mis-classified as intersecting with the roads that they pass under or over. See this figure for an example,  https://en.wikipedia.org/wiki/Spaghetti_Junction,_Birmingham

I think the word ‘fully’ should be removed.

Author Response

Dear reviewer,

thank you for providing feedback on our work. Please find the individual point-by-point responses below:

Line 81: The acronym CNN is used for the first time here but without being expanded. Acronyms should be expanded at first use, and the acronym placed in brackets.

Done

Line 84: The acronym NDVI is used for the first time here but without being expanded. Acronyms should be expanded at first use, and the acronym placed in brackets.

Done

Line 139: The paper mentions that Landsat image composites were utilised, however there is no mention of whether it was Landsat 8, 7, or older satellites. Given that each Landsat satellite is very different from previous one, it would be helpful to mention which one was the source of the image composites.

We added the information that we are using composites from all Landsat missions, as per reference (doi:10.1080/17538947.2016.1187673):

This approach uses the free and open-access Landsat archive [22] to produce yearly, gap-free, surface-reflectance image composites at a spatial resolution of 30 meters [23], taken from all available Landsat TM (Thematic Mapper) and ETM+ (Enhanced Thematic Mapper Plus) images.

Line 317: The acronym NBR is used for the first time here but without being expanded. Acronyms should be expanded at first use, and the acronym placed in brackets.

Done

Line 324: This line says the time series is shown in Figure 4 but I think it is shown in Figure 5.

Changed to Figure 6 in the updated manuscript.

Line 446-452: This is a minor comment and can be ignored. The paragraph describes an observation made regarding when the detected roads were built. It would be great if the authors could add one or two sentences stated what the observations suggest about the roads or the machine learning model.

Lines 503-508 in the discussion give an interpretation on the observations:

The histogram of road ages shows two peaks, one around 1984/5 and another one around 2005. The former peak is related to the change detection algorithm, where a decreasing NDVI or NBR in the first years of observation will result in the first year being detected as the year of change (cf. Fig. 5). The second peak in 2005 is not explainable by the algorithm alone, and further investigations requiring additional reference data (e.g., GPS tracks of logging trucks) should be carried out in the future.

Line 485: The paper suggests that the model could be used for development of a fully connected road network.

The word ‘fully’ is an over-stretch, given that the road extraction assumes that all road segments topologically intersect. Roads that pass under or over other roads could be mis-classified as intersecting with the roads that they pass under or over. See this figure for an example, https://en.wikipedia.org/wiki/Spaghetti_Junction,_Birmingham

I think the word ‘fully’ should be removed.

We removed the word 'fully' and replaced it with 'topological'. Note that our application is focused on rural forestry roads where over/underpasses rarely occur. We acknowledge this limitation, however, by adding a sentence in the discussion:

Note that we do not detect any over- or underpasses as disconnected parts, which would need to be manually corrected in the road network. However, these rarely occur in rural forest road settings.

Reviewer 2 Report

Comments and Suggestions for Authors

This paper is progressing well and is worth publishing.

Please review the following for my general remarks and suggestions:

- It is recommended to include the contribution of the research at the end of the abstract at line 28.

- It would be beneficial if the authors expanded the related works in section 1.

- It is recommended to add a paragraph at the end of the introduction section which demonstrates the layout of the paper stating the following sections and providing a brief explanation of each subsection. This will give a chance reader to have an overview of the remainder of the paper.

- it is suggested to improve the quality of the conclusion section. Adding another section at the end of the paper after the conclusion which discusses the limitations of the current study as well as presents some suggestions for future research is highly recommended.

Author Response

Dear reviewer,

thank you for providing feedback on our work. Please find the individual point-by-point responses below:

- It is recommended to include the contribution of the research at the end of the abstract at line 28.

We added our research contribution to the abstract (albeit not at the end, but in line 19):

Our research examines the potential of a pre-trained neural network (VGG-16 trained on ImageNet) transferred to the remote sensing domain. The classification is refined through post-processing, which considers spatial misalignment and road width variability.

We also add a sentence regarding road age to the end of the abstract:

Finally, we use the detected road locations to investigate road age by accessing an archive of Landsat data, allowing spatiotemporal modelling of road access to remote areas. This provides us important information e.g., for habitat modeling.

- It would be beneficial if the authors expanded the related works in section 1.

We expanded the related work section and added these papers:

Çalışkan and Sevim [17] presented an investigation of forest roads from 10 cm orthophotos using four different deep learning models. They found significant differences between the models, with overall accuracies ranging between 89.15% and 98.49% and Intersection over Union (IoU) ranging between 61.81% and 67.31%. Depending on the metric, different models performed better than the others.

Zhang and Hu [18] used a VGG-16 based approach to extract primary and secondary roads from high spatial resolution imagery (40 cm) of Canadian forests. While they preprocessed data using a Laplacian of Gaussian (LoG) operation to locate road candidates, they did not employ weights from a pre-trained neural network.

While approaches using coarser spatial resolution data acquired by satellites have been presented, they typically do not focus on forest roads. For example, Hormese and Saravanan [19] show how urban roads can be detected using the VGG-16 architecture. In a recent contribution, Kelesakis et al. [20] demonstrated the necessity of rural road data for a range of applications under different deep learning solutions.

17. Çalışkan, E.; Sevim, Y. Forest Road Detection Using Deep Learning Models. Geocarto International 2022, 37, 5875–5890, doi:10.1080/10106049.2021.1926555.

18. Zhang, W.; Hu, B. Forest Roads Extraction through a Convolution Neural Network Aided Method. International Journal of Remote Sensing 2021, 42, 2706–2721, doi:10.1080/01431161.2020.1862438.

19. Hormese, J.; Saravanan, C. A Comparison of Convolutional Neural Network Architectures for Road Classification from Satellite Images. In Proceedings of the 2018 International Conference on Inventive Research in Computing Applications (ICIRCA); July 2018; pp. 354–359.

20. Kelesakis, D.; Marthoglou, K.; Grammalidis, N.; Daras, P.; Tsiros, E.; Karteris, A.; Stergiadou, A. A Holistic Framework for Forestry and Rural Road Detection Based on Satellite Imagery and Deep Semantic Segmentation. In Proceedings of the Ninth International Conference on Remote Sensing and Geoinformation of the Environment (RSCy2023); SPIE, September 21 2023; Vol. 12786, pp. 126–134.

- It is recommended to add a paragraph at the end of the introduction section which demonstrates the layout of the paper stating the following sections and providing a brief explanation of each subsection. This will give a chance reader to have an overview of the remainder of the paper.

As we follow the MDPI/Remote Sensing standard guidelines for paper layout (i.e, Introduction, Materials and Methods, Results, Discussion, Conclusion), we opt to leave out an explanation of these sections.

- it is suggested to improve the quality of the conclusion section. Adding another section at the end of the paper after the conclusion which discusses the limitations of the current study as well as presents some suggestions for future research is highly recommended.

We have reworked the conclusion section and now provide an additional paragraph specifically discussing the limitations of the approach:

While overall satisfactory and use-case appropriate results were obtained, the presented method has certain limitations. First, the roads need to be visible in the satellite imagery to be detected, and canopy cover over road segments can hinder accurate detection. A clear contrast of the road surface to the surrounding areas must occur, which may prove problematic in clear-cuts and similar settings. In the road detection, the CNN sometimes failed to detect linear features well. Larger patches, i.e., larger fields of view of the neural network, could help to resolve ambiguities regarding these linear structures. With different network architectures having a larger field of view or context with which to learn, better results may be obtained.

Reviewer 3 Report

Comments and Suggestions for Authors

Author employ CubeSat Imagery from the Planet onstellation to predict the occurrence of road probabilities using a SegNet Convolutional Neural Network. The classification is refined through post-processing, which considers spatial misalignment and road width variability. This work is very meaningful and the overall quality of the paper is good, but there are the following questions:

1. It is difficult to effectively capture this paper's innovation in the introduction. If we only use some current methods, it is difficult to support the innovation of this paper. Suggest supplementing the contribution of this paper.

2. Lack of comparative methods to support the advancement of their methods.

3. The author should introduce the process idea of each step of this paper in detail with the flow chart and put forward the innovative part.

4. How to ensure the effectiveness of domain adaptation methods in new application neighborhoods in line 190.

5. When there are interrupted (discontinuous) roads in the road segmentation results, can they be solved in post-processing and how to solve them.

Author Response

Dear reviewer,

thank you for providing feedback on our work. Please find the individual point-by-point responses below:

1.It is difficult to effectively capture this paper's innovation in the introduction. If we only use some current methods, it is difficult to support the innovation of this paper. Suggest supplementing the contribution of this paper.

We have refined the innovation part in the introduction by adding some literature on the state-of-the-art (also see response to Reviewer #2). The last paragraph of the introduction has been revised to better portray the contribution of this paper.

  1. Lack of comparative methods to support the advancement of their methods.

While we do not provide a comparison to non-deep-learning or other deep-learning architectures (also see Reviewer 4), we believe that our contribution is very valuable for the community for the following reasons:

  • To the best of our knowledge, CNN approaches have not been used to extract forest road information from very high-spatial resolution satellite imagery (i.e., 3 m Planet images), so this represents a novel contribution given forest roads are typically no wider than 2-3 pixels.
  • Comparison between CNN approaches typically suffer from choice of study area and set-up of training, test, and validation sets, especially when doing spatial splits to avoid issues with spatial autocorrelation. Furthermore, the lack of high-quality training data hinders comparisons.
  • We present how raster road data can be automatically vectorized, and used to build a topological network, and how the detected road locations can be utilized for further analyses, e.g. the determination of road age for habitat modelling.

While we agree that a comparison of different network architectures may provide additional information on which CNN to use, we think that our manuscript already encompasses a significant amount of novelty and innovation (now also put forth by the flowchart) - and consider that additional analysis out of scope.

  1. The author should introduce the process idea of each step of this paper in detail with the flow chart and put forward the innovative part.

We have added a flow chart (now Fig. 2) to highlight the innovations in our study.

  1. How to ensure the effectiveness of domain adaptation methods in new application neighborhoods in line 190.

Domain adaptation effectiveness has been shown for many remote sensing applications. We added some considerations to the manuscript:

Such domain transfers have previously been shown to be efficient with satellite imagery (e.g., [37,38]).

37. Zou, M.; Zhong, Y. Transfer Learning for Classification of Optical Satellite Image. Sens Imaging 2018, 19, 6, doi:10.1007/s11220-018-0191-1.

38. Liang, Y.; Monteiro, S.T.; Saber, E.S. Transfer Learning for High Resolution Aerial Image Classification. In Proceedings of the 2016 IEEE Applied Imagery Pattern Recognition Workshop (AIPR); October 2016; pp. 1–8.

  1. When there are interrupted (discontinuous) roads in the road segmentation results, can they be solved in post-processing and how to solve them.

We have added some considerations to that end in the discussion:

As shown in Figures 7-9, the CNN sometimes fails to detect parts of the road, resulting in gaps in the binary road layer. One of the advantages of VecNet as a vectorization method is that small gaps – typically up to 50 m – can be bridged, as long as the detected probability does not drop too low and the gap does not occur at a tight turn [21]. Finally, if a full network is required, manual correction and editing can be carried out using standard GIS software.

Reviewer 4 Report

Comments and Suggestions for Authors

This paper presents a CNN-based approach for forest road network extraction from remote sensing image. Although the topic of this study on extraction of road network in rural areas is interesting and popular in the literature, there are some serious issues that require further explanation and clarification.

1) The review of current related work is very simple and inadequate. With the popularity of deep learning in NLP, classification and prediction, many deep learning models are being widely used for road extraction from remote sensing image, and the CNN-based model has become a common framework for object classification and segmentation in remote sensing images. For the presented paper, there is a lack of discussion on the performance and applicability of existing models in the application of rural road extraction. So, it is difficult for me to evaluate the advantages and innovations of the method proposed in this paper compared with current representative models. In addition, the research purpose of this paper and the problems to be solved are not very clear.

2) Different from urban areas, some mountainous areas or densely vegetated areas may be always covered by clouds and fog due to the influence of regional climate. Satellite remote sensing images are easily obscured by clouds and forest vegetation, resulting in information loss. Therefore, it is difficult to obtain useful road information from these image data. This paper lacks discussion and analysis on the treatment of such problems.

3) The details of the acquisition and processing of training datasets are missing.

4) In the presented paper, the authors stated that the performance of the residual U-Net neural network in remote, wooded landscapes is inferior, which is subjective and unproven. Based on my experience and knowledge, I think if the residual U-Net based model is well trained based on the forest road network data and it can obtain competitive results. In recent years, there are already some deep learning models have been used for road extraction in rural area from remote sensing image. The SegNet used in the paper has a simple structure and lacks consideration of different scale characteristics. Compared with other deep learning models (such as Residual U-Net and RoadNet), it lacks essential innovative structure or features. And different deep learning models may have different structural characteristics, so they are suitable for different data scenes and recognition tasks. The reason for using the SegNet rather than others should be discussed, and whether the SegNet is suitable for the data scene and the task of this study.

5) In the Vectorization steps, the VecNet is used, but how it is used is not clear.

6) According to the experimental results, the road extracted by the proposed method has some incomplete or broken parts, as shown in the upper right corner of Figure 6. For the experiments, it is suggested to add the comparison with results of state-of-the art models to highlight the superiority of the proposed method.

Author Response

Dear reviewer,

thank you for providing feedback on our work. Please find the individual point-by-point responses below:

1) The review of current related work is very simple and inadequate. With the popularity of deep learning in NLP, classification and prediction, many deep learning models are being widely used for road extraction from remote sensing image, and the CNN-based model has become a common framework for object classification and segmentation in remote sensing images. For the presented paper, there is a lack of discussion on the performance and applicability of existing models in the application of rural road extraction. So, it is difficult for me to evaluate the advantages and innovations of the method proposed in this paper compared with current representative models. In addition, the research purpose of this paper and the problems to be solved are not very clear.

We are fully aware of the multitude of CNN/DL architectures available and used for road detection. However, we focus on rural roads in forest settings using very-high-resolution satellite imagery (3 m pixel size). Most literature to date either aims to detect roads in urban settings or using imagery acquired from UAV or airborne platforms with pixel sizes of 0.05 - 0.4 m. In our use case, roads are typically 1-3 pixels wide only.

We have added - also in response to Reviewer #2 - the following references to the introduction to make the unique contribution of the paper more clear:

Çalışkan and Sevim [17] presented an investigation of forest roads from 10 cm orthophotos using four different deep learning models. They found significant differences between the models, with overall accuracies from 89.15% to 98.49% and Intersection over Union (IoU) ranging between 61.81% and 67.31%. Depending on the metric, different models performed better than the others.

Zhang and Hu [18] used a VGG-16 based approach to extract primary and secondary roads from high spatial resolution imagery (40 cm) of Canadian forests. While they preprocessed data using a Laplacian of Gaussian (LoG) operation to locate road candidates, they did not employ weights from a pre-trained neural network.

While approaches using coarser resolution data acquired by satellites have been presented, they typically do not focus on forest roads. For example, Hormese and Saravanan [19] show how urban roads can be detected using the VGG-16 architecture. In a recent contribution, Kelesakis et al. [20] present the necessity of rural road data for different applications and anticipate different deep learning solutions.

17. Çalışkan, E.; Sevim, Y. Forest Road Detection Using Deep Learning Models. Geocarto International 2022, 37, 5875–5890, doi:10.1080/10106049.2021.1926555.

18. Zhang, W.; Hu, B. Forest Roads Extraction through a Convolution Neural Network Aided Method. International Journal of Remote Sensing 2021, 42, 2706–2721, doi:10.1080/01431161.2020.1862438.

19. Hormese, J.; Saravanan, C. A Comparison of Convolutional Neural Network Architectures for Road Classification from Satellite Images. In Proceedings of the 2018 International Conference on Inventive Research in Computing Applications (ICIRCA); July 2018; pp. 354–359.

20. Kelesakis, D.; Marthoglou, K.; Grammalidis, N.; Daras, P.; Tsiros, E.; Karteris, A.; Stergiadou, A. A Holistic Framework for Forestry and Rural Road Detection Based on Satellite Imagery and Deep Semantic Segmentation. In Proceedings of the Ninth International Conference on Remote Sensing and Geoinformation of the Environment (RSCy2023); SPIE, September 21 2023; Vol. 12786, pp. 126–134.

2) Different from urban areas, some mountainous areas or densely vegetated areas may be always covered by clouds and fog due to the influence of regional climate. Satellite remote sensing images are easily obscured by clouds and forest vegetation, resulting in information loss. Therefore, it is difficult to obtain useful road information from these image data. This paper lacks discussion and analysis on the treatment of such problems.

By using composites of images acquired over a full year, the mentioned problem did not occur. We acknowledge that in other geographical areas this may be difficult, and we have added some information to the limitations:

While the geographic area chosen for our study did not cause any issues with permanent cloud-cover, we acknowledge that may be different for other areas and temporal periods. In the case of the Canadian boreal, forest fires have been increasing in magnitude and duration in recent years, which may cause atmospheric distortions. A successful detection of the roads using RGB satellite data under these conditions would not be possible.

3) The details of the acquisition and processing of training datasets are missing.

The training dataset is described in Section 2.4. We added some more information on the processing of this dataset for use with SegNet:

The road data was vectorized using GDAL [31], taking care to align the resulting raster with the RGB imagery. Pixels that were intersecting any of the filtered roads were set to value “1”, while the background was set to value “0”. To emulate road width representation in the binary training data, we morphologically dilated the data. We obtained road widths of 2 px or 6 m, which are typical values for the roads in our study area.

4) In the presented paper, the authors stated that the performance of the residual U-Net neural network in remote, wooded landscapes is inferior, which is subjective and unproven. Based on my experience and knowledge, I think if the residual U-Net based model is well trained based on the forest road network data and it can obtain competitive results. In recent years, there are already some deep learning models have been used for road extraction in rural area from remote sensing image. The SegNet used in the paper has a simple structure and lacks consideration of different scale characteristics. Compared with other deep learning models (such as Residual U-Net and RoadNet), it lacks essential innovative structure or features. And different deep learning models may have different structural characteristics, so they are suitable for different data scenes and recognition tasks. The reason for using the SegNet rather than others should be discussed, and whether the SegNet is suitable for the data scene and the task of this study.

The choice of the word "inferior" relates to the result Microsoft presented with their RoadsDetection dataset, as manual interpretation of the dataset has shown that it fails to represent most rural roads in the study set. We understand the confusion and have rephrased this:

A recent example is Microsoft’s RoadDetections dataset, using a Residual U-Net neural network [16] on Bing Imagery. By visual inspection of the study area in the dataset provided by Microsoft (https://github.com/microsoft/RoadDetections, last access 19-02-2024), we found that many of the roads were missing, which we attribute to the network mainly having been trained on paved roads in urban environments

Regarding the choice of SegNet over a residual U-Net, the choice for a simpler (non-residual) network was motivated by two facts: (1) preliminary experiments on a different detection task (deadwood detection) showed that when the object of interest is only a few pixels wide, ResNet did not provide significantly better results and (2) SegNet provides a simpler model, which may be easier to understand for users that are not as familiar with neural networks.

Finally, we emphasize that we did not do a study on the use of different CNN architectures, but rather show (a) the feasibility and (b) an end-to-end method for detecting forest roads of a few pixel width in very-high resolution satellite imagery.

5) In the Vectorization steps, the VecNet is used, but how it is used is not clear.

We have amended the information on how VecNet is used in Section 2.7:

For certain applications, including navigation purposes, a vectorized and topologically intact road network is required. To automatically derive such a network from satellite imagery, we use the VecNet approach outlined by [21]. This approach operates on the probability raster outputted by the CNN. Note that, despite the name, VecNet is not a neural network but a vectorization method to create road networks.

The vectorization employs a “driving” approach, where roads are followed along the most likely path, with intersections and turnoffs recoded to be driven at a later point. To start this algorithm, input segments (“seeds”) are required. Seeds are derived from the reference network by selecting the first and the last segment from each of the line elements. We assume that road networks will be connected, i.e., there will be no ‘islands’ of roads within the analysis area that are not connected to roads leading into the area.

Due to single patches not being able to predict roads correctly, gaps in the raster probability map appear. We tuned the parameters of VecNet such that it was able to only leap over small gaps in the road network.

On the large area of interest, we first merged the predictions for the individual overpasses by averaging the probabilities in overlapping areas. Subsequently, we vectorized roads by using the full existing road network layer as seeds for the algorithm. This follows the application of a road update focussing on additionally built roads, as we assume that even discontinued roads have a long-lasting spectral footprint. In the results, we therefore only report on additional roads being detected, with the caveat that these roads may also be false positives (as described above).

6) According to the experimental results, the road extracted by the proposed method has some incomplete or broken parts, as shown in the upper right corner of Figure 6. For the experiments, it is suggested to add the comparison with results of state-of-the art models to highlight the superiority of the proposed method.

We have added the reasons for these gaps as well as possible solutions to the relevant sections of the discussion and conclusion sections. A comparison with other DL methods, or with edge-detection-based methods is out of scope for this paper.

Round 2

Reviewer 3 Report

Comments and Suggestions for Authors

The author has made revisions to the paper as required and is recommended for acceptance.

Author Response

Thank you for reviewing our work!

Reviewer 4 Report

Comments and Suggestions for Authors

The revised paper still lacks some necessary details about some key steps in the proposed/used method, such as how the image data is procossed to generate the train samples, how to obtian the benchmark labels and how to train the SegNet. The post-processing (Vectorization) has the same problem.

The experimental results are not satisfactory and lack of comparison with the results of related algorithms. Thus, it is hard to demostrate the advantages and innovative of the proposed method. 

Author Response

Dear reviewer,

thank you for taking the time to review our manuscript again. We apologize for not fully fulfilling the requests you made in the previous iteration and have taken great care to be more clear in our explanations in this version. Please find our responses below:

The revised paper still lacks some necessary details about some key steps in the proposed/used method, such as how the image data is procossed to generate the train samples, how to obtian the benchmark labels and how to train the SegNet. The post-processing (Vectorization) has the same problem.

We have reworked the section on data preprocessing and label generation. For additional clarity, we have created a subsection "2.7 Dataset preparation" covering these topics. Parts of the content of this section are copied from previous sections 2.4 and 2.6, where we had covered these topics.

2.7 Dataset preparation

As the network architecture is limited to processing images of input size 244 × 244 pixels, patches of that size are created from the satellite imagery. To minimize edge effects, we create these patches with an overlap of 50%, i.e., 122 pixels. The patches are created on-the-fly from the larger raster datasets by the data loader.

For training and loss calculation, we use these patches, but when inferring the road network on the validation and test areas, we combine overlapping patches by a weighted average. Weights are calculated by a piecewise linear function (in x- and y-dimension) where weights are decreasing from the center, where the maximum is capped at ¼ of the patch width/height (Figure 4). The reasoning here is that the CNN is able to make better predictions in the image center, as the neighbourhood context is clearer. Thus, each pixel’s road probability is the result of four passes through the network. To obtain a final binary map for pixel-wise comparison, simple thresholding with a cut-off value of 0.5 is used.

The binary reference dataset (the labels) for both training and evaluation (Section 2.8) is created by using the BC provincial road database (Section 2.4). The road data is vectorized using GDAL [40], taking care to align the resulting raster with the RGB imagery. Pixels that intersect any of the filtered roads is set to value “1”, while the background is set to value “0”. To emulate road width representation in the binary training data, we morphologically dilate the data. We obtain road widths of 2 px or 6 m, which are typical values for the roads in our study area.

In the training process, we limit input patches to ones that contain at least 10 pixels classified as roads. This data-pre-selection speeds up training significantly, as patches with no roads present provide little information on how to detect roads. For validation and testing, we run the network on the full data, as no a-priori information on road presence is assumed.

Regarding training the SegNet, the training hyperparameters are given in Tab. 1. Details on the training can be found in the accompanying source code, if required.

For Section 2.9 Vectorization, we improved the language, hoping to add more clarity on how we used the method published by Roussel et al. (2023):

2.9. Vectorization

For certain applications, including navigation purposes, a vectorized and topologically intact road network is required. To automatically derive such a network from satellite imagery, we use the VecNet approach outlined by [21]. This approach operates on the probability raster outputted by the CNN. Note that, despite the name, VecNet is not a neural network but a vectorization method to create road networks.

The vectorization employs a “driving” approach, where roads are followed along the most likely path, with intersections and turnoffs recoded to be driven at a later point. To start this algorithm, input segments (“seeds”) are required. Seeds are derived from the reference network by selecting the first and the last segment from each of the line elements. We assume that road networks will be connected, i.e., there will be no ‘islands’ of roads within the analysis area that are not connected to roads leading into the area.

Using these seeds and the probability map generated by the CNN, a path of highest probability is followed. When an intersection is detected, additional seeds are created to allow following of multiple turnoffs. The vectorization uses the concept of friction of distance, where the cost (represented by the change in probability from one cell to another) is minimized. The algorithm processes the roads in small steps using a given window size to look ahead and find the most likely position of the roads. Additional constraints, such as a minimum probability and a maximum turning angle within the specified window, make the method more robust to false negative and false positive detections [21].

Due to single patches not being able to predict roads correctly, gaps in the raster probability map appear. We tuned the parameters of VecNet such that it was able to only leap over small gaps in the road network.

On the large area of interest, we first merged the predictions for the individual overpasses by averaging the probabilities in overlapping areas. Subsequently, we vectorized roads by using the full existing road network layer as seeds for the algorithm. This follows the application of a road update focussing on additionally built roads, as we assume that even discontinued roads have a long-lasting spectral footprint. In the results, we therefore only report on additional roads being detected, with the caveat that these roads may also be false positives (as described above).

The experimental results are not satisfactory and lack of comparison with the results of related algorithms. Thus, it is hard to demostrate the advantages and innovative of the proposed method. 

For our use-case, the results are satisfactory. We appreciate the call for comparison with related algorithms, but reiterate that we believe that a large-scale intercomparison of DL methods is out of scope for this paper. The innovation of our method lies therein that we show that it is (a) possible to detect narrow roads in satellite imagery of high spatial resolution and (b) these detections can be used to create either road networks or derive road age using auxiliary data.

We do not see our contribution as showing that SegNet is the best CNN for the task, but merely to show that CNNs like SegNet are able to perform the task, and that pre-training of the neural network using e.g., ImageNet data is beneficial.

Round 3

Reviewer 4 Report

Comments and Suggestions for Authors

No more questions.